# On Blockchain-Enhanced Secure Data Storage and Sharing in Vehicular Edge Computing Networks

Muhammad Firdaus [1] and Kyung-Hyune Rhee [2,*]

1   Departement of Interdisciplinary Graduate Program of Artificial Intelligence on Computer, Electronic and Mechanical Engineering, Pukyong National University, Busan 48513, Korea; mfirdaus@pukyong.ac.kr
2   Departement of IT Convergence and Application Engineering, Pukyong National University, Busan 48513, Korea
*   Correspondence: khrhee@pknu.ac.kr

**Abstract:** The conventional architecture of vehicular ad hoc networks (VANETs) with a centralized approach has difficulty overcoming the increasing complexity of intelligent transportation system (ITS) applications as well as challenges in providing large amounts of data storage, trust management, and information security. Therefore, vehicular edge computing networks (VECNets) have emerged to provide massive storage resources with powerful computing on network edges. However, a centralized server in VECNets is insufficient due to potential data leakage and security risks as it can still allow a single point of failure (SPoF). We propose consortium blockchain and smart contracts to ensure a trustworthy environment for secure data storage and sharing in the system to address these challenges. Practical byzantine fault tolerance (PBFT) is utilized because it is suitable for consortium blockchain to audit publicly, store data sharing, and records the whole consensus process. It can defend against system failures with or without symptoms to reach an agreement among consensus participants. Furthermore, we use an incentive mechanism to motivate the vehicle to contribute and honestly share their data. The simulation results satisfy the proposed model's design goals by increasing vehicular networks' performance in general.

**Keywords:** Blockchain; smart contracts; privacy and security; PBFT; incentive mechanism; vehicular edge computing

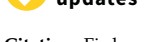



## 1. Introduction

With a centralized approach, the traditional architecture of vehicular ad hoc networks (VANETs) has trouble overcoming the increasing complexity of applications for intelligent transportation systems (ITS) accompanied by difficulties in the provision of large volumes of data storage, trust management, and information security. Therefore, the concept of vehicular edge computing networks (VECNets) has evolved to be a promising paradigm that brings multiple opportunities to support computing, sharing, and massive data storage close to vehicles as a data provider by offering real-time transaction processing [1]. VECNets are defined as the integration of mobile edge computing with vehicular networks. It aims to decrease the overhead by placing computing resources closer to the network's end users. However, security and privacy issues are critical challenges for VECNets due to their risk of data leakage and single point of failure (SPoF) in the centralized server approach. Therefore, it is necessary to design a secure data management system in VECNets without an intermediary or a centralized approach.

On the other hand, the concern of data privacy issues increases significantly in the context of data storage [2] and data sharing in a vehicular network because VECNets contain sensitive information of users, such as driving preferences, customer identities, and vehicle numbers. Therefore, the participant may be reluctant to store and share their data in the system due to the risk of various malicious activities that might jeopardize the system's security and participants' privacy for personal benefit. Ideally, all data are sent

anonymously to overcome this issue as the data management environment is still untrusted. Therefore, the real identity of the user cannot be known by other users. However, even though it can solve the risk of a violation of privacy, it does not guarantee the data's reliability. Moreover, the user's interest will be reduced as the users do not benefit when they share their data or contribute to assessing the data's integrity from other users [3]. Therefore, the incentives mechanism is can be leveraged to encourage users or vehicles to store and share their data while assessing the credibility of data in VECNets.

Since Nakamoto introduced Bitcoin [4], in the past ten years blockchain has gained much popularity as an emerging technology to provide better security on data sharing among many parties without an intermediary. Blockchain is considered an appropriate solution to address the privacy security issue [5], which can facilitate a secure, trusted, and decentralized intelligent transportation system [6]. In this sense, by looking at the blockchain's merits, the centralized server can be replaced by adopting a decentralized blockchain approach. In recent years, several works have been widely studying blockchain-based implementation in vehicular networks [7]. For instance, blockchain is utilized to form a decentralized trust management system in vehicular networks by allowing the neighboring vehicle to validate the received data using the Bayesian inference model [8] for preventing inappropriate data from a malicious vehicle [9]. Besides, consortium blockchain with joint Proof-of-Work (PoW) and practical byzantine fault tolerance (PBFT) consensus mechanism is used in order to achieve secure data sharing and storage system [10]. Further, by exploiting its smart contracts, blockchain efficiently manages a vehicle's reputation [11] and provide incentives based on the calculation of vehicle reputation values [12].

Motivated by the above developments, in this paper, a consortium blockchain and smart contracts are utilized to form distributed data storage and secure data sharing in VECNets. A consortium blockchain is a specific blockchain where the multiple nodes are preselected [13] and authorized to execute the consensus process and determine the generation of each block. Here, RSUs are defined as a preselected node (chosen by the Department of Transportation and the consortium members) placed at the network's edge. Moreover, we use the PBFT algorithm for the consensus mechanism as it is more suitable for consortium blockchain [14]. PBFT is presented to audit publicly, store data shared, and record the whole consensus process. It can also defend against system failures with or without symptoms to reach an agreement among consensus participants. Furthermore, we allow the schema of the vehicle's data reputation based on its credibility assessment process to prevent irrelevant data information. Further, an incentive mechanism is proposed to motivate vehicles to contribute honestly and share their data to maintain and improve system reliability. In short, this paper's contribution can be summarized as follows.

1. We propose a decentralized trust system on vehicular edge computing network by utilizing a consortium blockchain and smart contracts.
2. We aim to enhance reputation-based data management's security using blockchain to encourage vehicles to contribute honestly to VECNets data sharing transactions.
3. We conduct the simulation to show that our proposed architecture increases the vehicular network's general parameter performance.

The remainder of this paper is organized as follows. In Section 2, we give background knowledge related to vehicular edge computing networks and blockchain technology. Then, we present the design architecture of blockchain-based VECNets, a secure and decentralized framework in vehicular networks, in Section 3. In Section 4, we discuss the relevant security feature and performance evaluation of our proposed system. Finally, Section 5 concludes this paper.

## 2. Background

Our work is closely related to vehicular edge computing networks and blockchain technology. We give background knowledge in this section.

### 2.1. Vehicular Edge Computing Networks

Vehicular edge computing networks (VECNETs) are an extended concept of the conventional vehicular ad hoc networks (VANETs), where an additional edge layer is the main difference that distinguishes between VANETs and VECNets. The idea of VECNets is to combine the conventional vehicular network (VANET) and mobile edge computing (MEC) technology that was introduced and standardized by ETSI Industry Specification Group (ISG) in December 2014 [15]. MEC aims to enhance user experience with high bandwidth, low latency, and real-time communications [15]. Furthermore, it reduces delay and response time on decision-making computation by leveraging the edge node as local server infrastructure. Recent studies have been proposed for the integration between VANETs and MEC. In [16], the authors suggested a joint load balancing and offloading problem in VECNets to improve network effectiveness. The authors of [17] proposed DREAMS, an efficient distributed reputation management in VECNets. In [18], the authors introduced a new mechanism to reduce resource sharing cost while increasing quality of service (QoS) parameters with mixed-integer nonlinear programming (MINP) formulation.

Generally, the architecture of VECNets consists of a user layer, an edge layer, and a cloud layer [19,20]. In the user layer, vehicles are equipped with onboard units (OBUs) to communicate with the roadside unit (RSUs) in the edge layer to access network services. There are two types of communication in vehicular networks: vehicle to vehicle (V2V) and vehicle to infrastructure (V2I). In the case of V2V communications, vehicles communicate with a neighboring vehicle using OBUs. OBUs are equipped with simple computation capabilities to collect data from sensing devices. Then, V2I, which consists of OBUs and RSUs, establishes a connection with the help of dedicated short-range communication (DSRC) in a single or multi-hop communication [21]. Here, RSUs are deployed along the road acts as an edge node, which provides wireless communications from roadside infrastructure to vehicles [22]. Next, the edge layer consists of several nearby RSUs responsible for temporary data storage that then deliver the data periodically to the cloud layer. A central cloud server is a permanent data center that can provide massive data storage and execute complicated computing tasks in the cloud layer. This server manages all the edge nodes to offers the service in VECNets.

### 2.2. Consortium Blockchain

In 2008, Nakamoto proposed a peer-to-peer (P2P) digital currency system named Bitcoin for economic transactions based on blockchain technology [4]. Blockchain is a distributed ledger technology that allows participant nodes to share and approve transactions in the network. Then, those transactions that contain data recorded with time-stamp are validated by the particular consensus mechanism before it is stored in an immutable database. Generally, blockchain has three functions: realizing decentralized storing, acting as a distributed ledger, and supporting distributed services relying on smart contracts [23]. Currently, several researchers from different fields are attracted to develop blockchain due to its advantages. Several advantages of blockchain are described as follows:

- **Anonymity and privacy**: Blockchain allows the user to join the network anonymously. Therefore, the real identity of the user cannot be known by other users. Thus, blockchain is able to provide privacy to the user.
- **Immutability**: This feature is one of the essential blockchain features, where the stored data cannot be deleted or modified from the network as the data was recorded and confirmed in the blockchain. Thus, blockchain makes data difficult to be added or altered arbitrarily [19].
- **Decentralized**: The core value of blockchain is to enable a database to be directly shareable without a central server. The characteristic of a decentralized approach

shows the fragmentation of control over the whole system. It allows the system to achieve open participation, immunity from particular attacks, and the elimination of single points of failure [24]. Consequently, the blockchain data are consistent, reliable, have time-stamping for recorded transactions, and widely accessible.

- **Transparency**: Every user can access the transactions transparently once data is stored in the blockchain. Therefore, all the stored data in the blockchain are publicly viewable by the user in the network.
- **Distributed and trustful transaction**: In the blockchain, every node can share and validate the transactions in a distributed manner without the control of the central trusted authority.

According to the access restriction to the network and node participating in the consensus process, blockchain is distinguished into three types: a public blockchain, private blockchain, and consortium blockchain. A public blockchain is an open blockchain, where anyone can join the network and contribute to a consensus mechanism without permission from a trusted authority. By contrast, private blockchain gives full control to a trusted party for managing participants' access to the network service while selecting a few participants who can conduct the consensus process [25]. The latter, consortium blockchain, provides the same benefits as the private blockchain, but without reliance on a trusted party. A consortium blockchain is a specific blockchain that allows the preselected nodes to execute the consensus process with a particular mechanism [26].

### 2.3. Other Preliminaries

### 2.3.1. Smart Contracts

The concept of smart contracts was introduced in the 1990s by cryptographer and computer scientist Nick Szabo [27]. A smart contract is a self-executable computer protocol that can digitally verify, simplify, and accomplish the contracts made between parties on blockchain. Smart contracts facilitate a secure transaction, which cannot be interrupted or modified because of the distributed approach's nature. Besides, smart contracts have several characteristics, such as verifiability, decentralization, and enforceability. This characteristic allows the smart contracts to be automatically executed between parties without supervision in a central server or a trusted authority [28]. Therefore, smart contracts could improve the efficiency, reliability, and privacy security of VECNets.

### 2.3.2. Consensus Algorithm

A consensus mechanism is a set of rules that is used to generate the block data value by achieving the necessary agreement among the participating nodes in the decentralized blockchain network. Here, the blockchain system should ensure that all transactions are trustworthy, along with the agreement of the participating node on the particular consensus algorithm implemented in the network. There is a variety of consensus algorithms in distributed systems, such as PoW, proof of stake (PoS), delegated PoS (DPoS), Casper, proof of elapsed time (PoET), and PBFT [29]. In order to provide knowledge about the consensus algorithm, Table 1 shows its comparison. In this paper, we use the PBFT algorithm as it is more suitable for consortium blockchain and it offers strong consistency for VECNets [14].

**Table 1.** Comparison of consensus algorithms.

| Algorithm | PoS | dPoS | Casper | PoET | PBFT |
|---|---|---|---|---|---|
| **Decentralized** | complete | complete | complete | semi | semi |
| **Performance** | relatively high | high | relatively high | high | high |
| **Technical maturity** | mature | mature | not-applied | not-applied | mature |
| **Malicious number** | 51% | 51% | 51% | 51% | 33% |
| **Blockchain type** | public | public | public | consortium | consortium |
| **Token** | yes | yes | yes | no | no |

### 2.3.3. Incentive Mechanism

The utilization of an incentive mechanism motivates vehicles to contribute and honestly share their data (i.e., traffic information) to improve the system's reliability. The existing scheme of incentive mechanisms such as the vehicle's reputation-based [30] and payment-based [31] generally use a central trusted authority. The trusted authority guarantees direct communication and secure transaction to prevent fraudulent vehicle behaviors. Therefore, the system's reliability, equity, and quality are influenced by the central trusted authority as a service provider. However, the centralized incentive mechanism is insufficient as it still facing the risk of a SPoF. It is worth noting that a single mistake affects the entire system orchestration. Further, the server may be a bottleneck caused by the necessity of large amounts of vehicle's transactions [32]. In the context of user privacy, a central service provider may also expose vehicles' private data or trade it for personal benefit. Once vehicles become malicious, they can manipulate the data sharing process by providing the wrong information to gather private information from other vehicles. Therefore, the privacy issue needs to be considered to achieve trustworthy and fairness VECNets. For this reason, we propose the decentralized incentive mechanism based on blockchain and smart contracts to achieve trustworthy environments. Our goal is to enhance reputation-based data management's security using blockchain to encourage vehicles to contribute honestly to VECNets data sharing transactions.

### 2.4. Related Work

Recently, there exist several works that focus on the study of blockchain in vehicular networks. In [7], the authors review the latest research activities of the blockchain-based vehicular environments by identifying research challenges and technical issues. They divide blockchain utilization into three layers: perception layer, networking layer, and application layer. First, the perception layer uses blockchain to solve the trust management issue and improve the perception accuracy on data sharing in the vehicular network. Second, the networking layer for network security and achieving a secure incentive mechanism solution. The last layer is the application layer for solving accountability, security, and privacy issues in a decentralized vehicular system.

In [8], blockchain is utilized to form a decentralized trust management system in vehicular networks by allowing the neighboring vehicle to validate the received data using the Bayesian inference model for preventing inappropriate data from a malicious vehicle. Here, RSUs perform as miners to generate the block by employing joint PoW and PoS consensus algorithm. Moreover, the authors of [9] also propose a Bayesian network implemented in RSUs, which aims to easily detect fake information by calculating the subsequent probability of an event, taking into account pre-traffic probability, traffic event period, vehicle integrity, and the event collected from the vehicle.

Additionally, to minimize attacks from malicious nodes in the consensus process, many works consider applying a consortium blockchain framework. In [10], the authors propose a consortium blockchain for a data sharing framework for the vehicular network to solve the risk of malicious tampering in a centralized data storage scheme. Here, they employ a smart contract in preselected nodes to manage data sharing and data storing in the system. The integrity and security of the data can be ensured by a digital signature technique when the vehicles upload their data. On the other hand, the block transactions are created and stored in the blockchain network by deploying a joint PoW and PBFT algorithm for its consensus mechanism. Moreover, the authors of [11] present a secure P2P data sharing system in VECNets by exploiting its smart contracts and applying the vehicle's reputation scheme for system efficiency. Here, they use joint PoW and PoS for consensus scheme and a three-weight subjective logic model for managing the vehicle's reputation. In order to motivate the vehicle node in providing accurate and timely data sharing, the authors of [12] propose a rewarding scheme using Ethereum smart contract incentives based on the calculation of vehicle reputation values. The authors propose

Proof-of-Authority (PoA) consensus mechanism to validate the transactions and enhance the system performance.

## 3. Proposed Architecture

In this section, we explain the design architecture of blockchain-based VECNets, a secure and decentralized framework in vehicular networks, as shown in Figure 1. This architecture use consortium blockchain with preselected RSUs, and they are legitimate to execute the process of consensus while providing a decentralized reputation-based incentive mechanism to ensure trusted transaction processing. Here, we define the entities used in this paper, describe the design architecture in each layer, and give each step's detailed procedures in the proposed system.

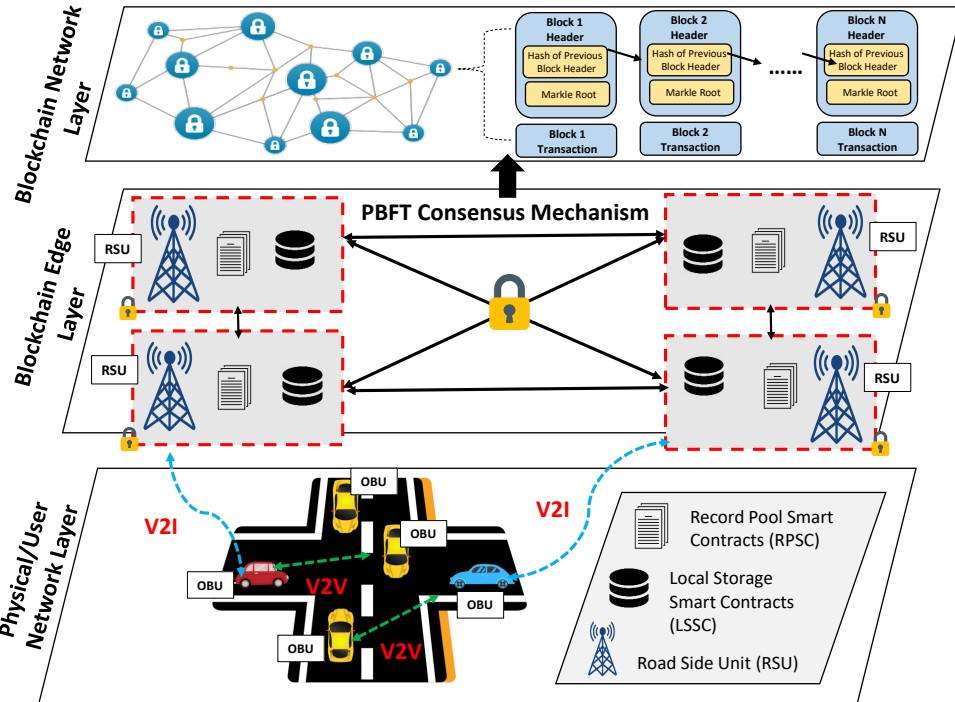

**Figure 1.** Overview of design architecture.

### 3.1. Entities of Proposed Scheme

Before introducing design architecture, we give definitions of related terms used in this paper.

- **Vehicle**: The vehicle is the primary entity of VECNets that contains OBUs, which are used to communicate with RSU or other vehicles. An OBU is equipped with a memory unit, a communication module, and a sensor device to make simple computation and communication, as well as collect detailed information. Each vehicle has public and private keys to share their data using V2V and V2I communication. Here, those keys are used for encrypting the information message that is generated by vehicles in VECNets.
- **Information message**: There are two categories of information message that are distinguished based on the message generated by the sending and receiving vehicle. The sending vehicle generates and broadcasts message information to the neighboring vehicle and nearest RSUs. This message contains report events, such as accident information, safety warnings, traffic jams, weather conditions, and snow reports on the road. Meanwhile, the receiving vehicle (neighboring vehicle) generates and uploads the information message to the RSUs that contain the trust value rating of the message information received from the sending vehicle.

- **Road side unit (RSU)**: RSUs represent the traffic handler that provides wireless communication from roadside infrastructure to the vehicles. RSUs are distributed along the road and are designed to manage a group of OBUs in a specific distance range. RSUs have higher computing power and storage compared to the OBUs. Therefore, RSUs have preselected nodes in consortium blockchain that participate as miners in the consensus mechanism process.
- **Local storage and record pool**: In this paper, smart contracts are exploited to achieve efficiency, reliability, and secure data storage and sharing in VECNets. Here, two smart contracts, i.e., local storage smart contracts (LSSC) and record pool smart contracts (RPSC), are deployed on distributed edge nodes (RSUs) to collect the trust value rating and conduct the consensus mechanism.
- **Trusted authority**: Trusted authority (TA) is responsible for managing the key, generating public parameters, and managing the users' identities.

### 3.2. Design Architecture of Proposed Scheme

The architecture of the proposed scheme consists of a physical/user layer, blockchain edge layer, and blockchain network layer as shown in Figure 1, with the function of each layer is described as follows.

- **Vehicle Network Layer**: This layer is responsible for vehicle registration and authentication, message broadcasting by the sending vehicle, and rating generation by the receiving vehicle. Before access the network, each vehicle should register to TA (e.g., Department of Transportation for vehicle management) to be an authorized vehicle. After passing the authentication process, vehicles with many sensors can automatically collect data using its sensing devices related to the occurring events on the road. With the help of OBUs, the vehicle broadcasts the information message at a specific time and location using V2V and V2I communication. However, the sender vehicle may act as a malicious vehicle that provides the wrong information. To cope with that issue, the neighboring vehicles at a certain distance and time close to events occurred are allowed to evaluate the received information. Then, they generate the rating based on the message's credibility, and they upload the rating to the nearest RSUs in the edge layer.
- **Blockchain Edge Node Layer**: This layer performs the critical process of calculating message credibility and the consensus mechanism. Here, RSUs are chosen to verify the data from the vehicle network layer. The sending vehicle periodically transmits the information message while the receiving vehicle uploads the evaluation result of the information message's trust value. Then, RSU is authorized to store the data to the blockchain network layer by performing the consensus mechanism. RSUs, as the blockchain edge nodes, contain LSSC and RPSC smart contracts. LSSC is responsible for collecting data uploaded by vehicles and receiving data shared by other RSUs in distributed frameworks. RPSC stores the data into a blockchain network layer using the PBFT consensus mechanism. The data collected by RPSC contains the result of the calculation trust value rating based on majority rule to become the candidate block in the consensus process.
- **Blockchain Network Layer**: After a consensus mechanism creates the new data block and is stored into the blockchain as a distributed ledger, the blockchain network provides a distributed incentive mechanism based on participants' contribution to maintain the system's sustainability.

### 3.3. Procedures of Proposed Scheme

Figure 2 explains the proposed system's procedures for secure data storage and sharing in VECNets based on a consortium blockchain. Table 2 summarizes the notation used to describe our proposed method. The details are described as follows.

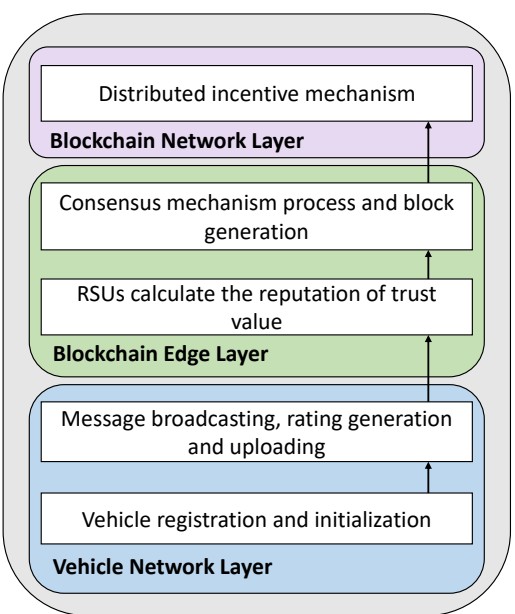

**Figure 2.** Procedures of proposed architecture.

**Table 2.** Summary of notations.

| Symbol | Description |
|---|---|
| $V_i$ | The $i$th vehicle in the network |
| TA | The trusted authority for vehicle registration |
| $ID_i$ | The $i$th identity of vehicle |
| $PK_ID, SK_ID$ | Public and private key pair of vehicle |
| $Cert_ID$ | The corresponding certificate of vehicle |
| $V_s$ | The sending vehicle which collect and broadcast the information message |
| $E_nMsg^p_{i,s}$ | The message encrypted by $V_s$ with a specific time $i$ and location $p$ |
| $V_r$ | The receiving vehicle (the neighboring vehicle) which generate the rating of message |
| $cb_{Msg^p_{i,r}}$ | The credibility of the message |
| $R^{s,r}_{Msg}$ | The message rating uploaded by $V_r$ |
| $\delta^{i,p}_s$ | The trust value rating aggregated by RSU |
| $Block_{data}$ | The candidate for a new block on the process of consensus mechanism |

### 3.3.1. Vehicle Registration and Initialization

Before joining the network, every vehicle needs to register to TA that grants the vehicle identity legitimation. After passing identity authentication, a legitimate vehicle ($V_i$) with true identity ($ID_i$) generates its public keys ($PK_ID$), private keys ($SK_ID$), and certificates ($Cert_ID$) to get access to the network service. Meanwhile, the cryptography algorithm (e.g., ECDSA) will guarantee the vehicle identity anonymity and communication security in VECNets.

### 3.3.2. Message Broadcasting, Rating Generation, and Uploading

Later, $V_i$ downloads the latest data from LSSC (local data storage of edge node) of nearby RSU on the system initialization step. Then, the sending vehicle $V_s$ collects the message information ($Msg_s$) using its sensing devices and broadcasts the message encrypted ($E_nMsg^p_{i,s}$) to the network with a specific time $i$ and location $p$. LSSC, as an RSU

local data storage, collects the broadcasted data from $V_s$. On the other hand, the neighboring vehicles, as the receiving vehicles ($V_r$), classify all message into groups ($M_i^1, M_i^2, ...M_i^p, ...$), then calculate and aggregate the credibility of the message ($cb_{Msg_{i,r}^p}$) based on Equation (1) as follows.

$$cb_{Msg_{i,r}^p} = \alpha + E_n Msg_{i,s}^{p\ -\beta.d_m}$$　　　　　　(1)

where $E_n Msg_{i,s}^p$ is an encrypted information message of $V_s$ that contains several related information, $E_n Msg_{i,s}^p = (Msg_s || timestamp || location)$. Both $\alpha$ and $\beta$ are predefine parameters, in which $\alpha$ represents the lower bound of message rating, whereas $\beta$ is a parameter that controls the rate of message credibility [8] based on $d_m$. Here, $d_m$ is the distance between $V_i$ and event location of $E_n Msg_{i,s}^p$. Using Equation (1), $V_r$ obtains a credibility set $C_s^{i,p}$ for $Msg_s$, where $C_s^{i,p} = (cb_{s,1}^{i,p}, cb_{s,2}^{i,p}, cb_{s,3}^{i,p}, ...cb_{s,r}^{i,p})$. $cb_{s,r}^{i,p}$ represents the credibility message of $V_s$, which is assessed by $V_r$. Correct messages generate a positive rating (+1); otherwise, a negative rating (−1). Next, $V_r$ uploads the message rating ($R_{Msg}^{s,r}$) of $V_s$ to nearby a RSU using the format $R_{Msg}^{s,r} = (V_s, V_r, E_n Msg_{i,s}^p, rating)$.

### 3.3.3. RSUs Calculate the Reputation of Trust Value

After the neighboring vehicles upload the message ratings $R_{Msg}^{s,r}$ into RPSC in edge layer, then the RSUs aggregate the trust value rating ($\delta_s^{i,p}$) of the message ($E_n Msg_{i,s}^p$) based on majority rule. Assuming that malicious vehicle cannot control most of the vehicle in the network, weighted aggregation is used to calculate trust value $\delta_s^{i,p}$ of the message as described in Equation (2).

$$\delta_s^{i,p} = \frac{\sum_{j=1}^r cb_{s,r}^{i,p} * R_{Msg}^{s,j}}{r}$$　　　　　　(2)

From Equation (2), the information message is credible only when the value of $\delta_s^{i,p}$ is greater than 0.5 as the majority rule's minimum threshold. Otherwise, the information message is considered untrustworthy, and the system will discard it. RPSC record all the participants $V_r$ that contribute to the trust value calculation based on their rating upload $R_{Msg}^{s,r}$. Further, the result of trust value $\delta_s^{i,p}$ becomes a candidate for a new block $Block_{data}$ on the consensus mechanism process.

### 3.3.4. Consensus Mechanism Process and Block Generation

In this step, only the authorized RSU performs the consensus process for storing authentication data and logs into the blockchain network layer. Figure 3 illustrates the consensus mechanism in the blockchain-based VECNets system. In this consensus process, PBFT is utilized due to its merits, such as consistency, maturity, high efficiency, and small resource consumption, making it suitable for our proposed scheme. Besides, it offers strong consistency for the consensus process among RSUs in VECNets [14]. Further, PBFT allows the existence of anomalous nodes ($f$), where $f = (n-1)/3$, without affecting the consensus result among the number of nodes ($n$) [33]. Figure 3 shows that the speaker $RSU_1$ broadcast $Block_{data}$ using a *pre-prepare* message to other authorized RSUs in edge node of consortium blockchain. Other RSUs ($RSU_2, RSU_3, ...RSU_n$) as congress members receive the data shared using its LSSC. Then, congress members verify the $Block_{data}$ and broadcast the *prepare* message among RSUs while calculating the message received from other congress members at the same $t$ interval. If the number of *prepare* messages received from different congress members are over $2f + 1$, they broadcast the *commit* among the participants (i.e., other RSUs). When receiving over $f + 1$ the *commit* message, the speaker confirms that the consensus is finished by generates a new block to the blockchain network. Consequently, the authentication $Block_{data}$ and logs are broadcast among node participants, and their ledger is updated. Otherwise, the block will be discarded, and the next round

consensus will be executed. Note that in Figure 3, $RSU_3$ represents the malicious node example that does not respond to the request from other nodes but still cannot influence the system decision.

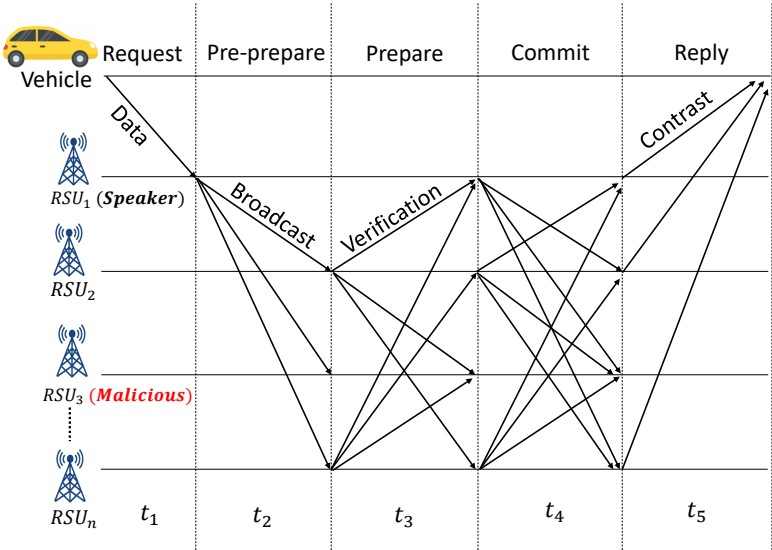

**Figure 3.** Consensus process in record pool smart contracts (RPSC).

### 3.3.5. Distributed Incentive Mechanism

After the consensus mechanism generates the new block, that block is automatically distributed to all local storage of RSU. The blockchain network provides an incentive mechanism based on the contribution of participants. The rewards are obtained by vehicles with a good rating that generate the correct messages and contribute to the trust value assessment. In Figure 4, we assume that $V_s$ is already known to provide a valid message about a particular event in the regional A with a specific time. $V_r$, as the neighboring vehicle on the same region, assesses the credibility message of $V_s$ by uploading the rating to the RSU. Later, RSU stores the correct message to the blockchain network. $V_s$ and $V_r$ obtain rewards based on their contributions in providing and assessing the information in a trusted manner. On the other case, for example, $V_q$, which is on the regional B, requests the specific data in regional A. $V_q$ chooses and downloads the message provided by $V_s$ based on the positive rating provided by $V_r$. Then, $V_q$ transfers payment for specific data requested to the system. Therefore, $V_s$ as data provider obtains an income from the system. Therefore, the system enables vehicles to download and forward packets for others securely.

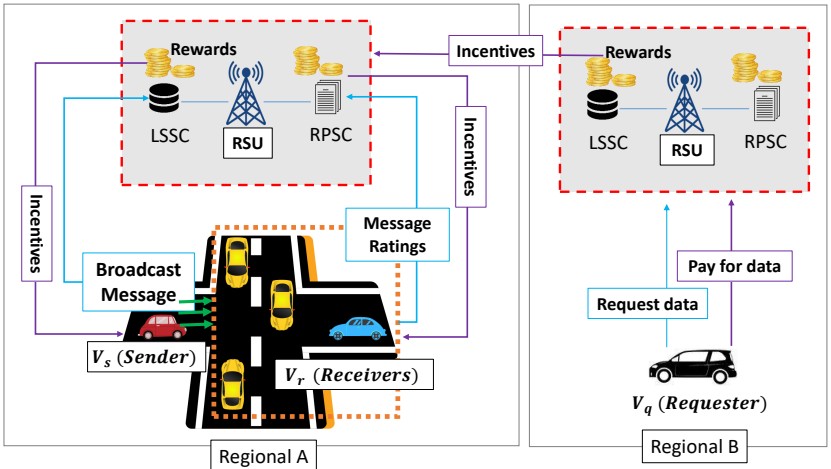

**Figure 4.** Illustration of incentive mechanism.

## 4. Security Analysis and Performance Evaluation

This section discusses the relevant security features and performance evaluation for our proposed system design to achieve decentralized trust management in VECNets. Overall, we discuss the advantages of decentralized management system. We conduct the simulation to implement our proposed architecture based on the performance parameters of vehicular networks.

### 4.1. Security Analysis

This paper aims to improve the security aspect for storing and sharing the data by leveraging blockchain technology, as described below.

#### 4.1.1. Decentralized Approach without an Intermediary

Compared to the traditional VECNets framework that employs the centralized approach, our proposed model leveraging consortium blockchain enables distributed edge nodes throughout the network. Every edge node (RSUs) has a blockchain replica that maintains a reliable database cooperatively. Therefore, vehicles can securely store and share their data without a central trusted party intermediary in a P2P manner. Furthermore, blockchain, as the model of a decentralized approach, effectively avoids a SPoF as the traditional attack of a centralized system. Thus, the vehicle can safely share its data without worrying that intermediary parties will breach it.

#### 4.1.2. Non-Repudiation and Data Consistency

Consortium blockchain selects RSUs to manage secure data sharing among authorized vehicles. In this sense, only legitimate vehicles can broadcast, receive, and assess the message information. Moreover, a cryptography technique ensures identity authentication during the data sharing process. All data transactions are recorded and publicly audited by distributed RSUs in the blockchain edge layer. Later, the consensus mechanism ensures the integrity of the data before appending it into the block. Thus, it is arduous to compromise all nodes in the entire network due to the immense cost (if doing so). On the other hand, the message reputation and consensus process can detect an error or incorrect data from malicious entities. Further, the PBFT consensus mechanism offers the system consistency that ensures all nodes have the same ledger simultaneously. Therefore, after the consensus mechanism generates and stores a new block to the blockchain network layer, a new block will be distributed to all edge nodes (i.e., RSUs). Thus, all legitimated vehicle in the network obtains updated information from RSU's local database.

#### 4.1.3. Tamper-Proofing and Data Unforgeability

In the vehicle network layer, the malicious vehicle cannot pretend to become a legitimate vehicle to corrupt the data because it cannot forge a digital signature that is encrypted by the vehicle's private key. In the blockchain edge node, the consortium blockchain provides honest RSUs selected to execute the consensus phase. Therefore, the small numbers of compromised RSUs will not affect the result of the consensus mechanism. Moreover, the PBFT consensus mechanism still works if there are 33% of malicious RSUs. Further, a few malicious RSUs cannot tamper with the stored block due to the chain structure that requires rebuilding the whole chain in the blockchain network layer.

#### 4.1.4. Data Availability

In our proposed model, the legitimate vehicles can securely download and forward packets in terms of data storing and sharing, which always available due to the distributed nature of blockchain. Moreover, they can communicate with the RSUs and request information on another vehicle's trustworthiness in the network. As shown in Figure 4, distributed RSUs collect a vehicle's reputation based on trust value ratings in the smart contract. Upon receiving a request message, the RSU verifies the identity of the requester's

vehicle. Consequently, RSU responds the request by sending the information of the target vehicle's reputation to the requester vehicle.

### 4.1.5. Data Privacy and Credibility

In order to prevent the attack from a malicious vehicle attempting to gather the vehicle's private data, we consider combining blockchain smart contracts and the cryptography algorithm that guarantees the vehicle identity anonymity and communication security in VECNets. We propose the vehicle's message reputation to tackle fake messages that may cause traffic congestion or even accidents. The broadcasted messages obtain ratings from its neighboring vehicle assessments to ensure the message's credibility. Moreover, our proposed scheme enables vehicles with a good rating as well as vehicles that contribute to message assessment to obtain certain rewards. Thus, they are motivated to be honest to share their data in the vehicular network.

### *4.2. Performance Evaluation*

#### 4.2.1. System Setup

We design a highway traffic scenario using the simulation of urban mobility (SUMO) supported by the OSMWebWizard package to prototype and evaluate our proposed model's performance. We use NS-3 as a discrete-event network simulator to analyze a vehicle mobility trace file to validate the result. The wireless access for the vehicular environments (WAVE) protocol defines the network according to the IEEE 802.11p @ 5.9 GHz standard compatible with DSRC communication among vehicles and RSU. We consider utilizing an optimized link-state routing protocol (OLSR) due to its efficiency based on delay, mobility, and speed in packet delivery, compared to other protocols [34]. Furthermore, we utilize Hyperledger Sawtooth [35], the new member of the Hyperledger family, to implement consortium blockchain in our scenario. We use Docker containers to run validators, transaction processors, and Sawtooth REST server, which in our case acts as the RSUs, smart contracts, and blockchain application interface, respectively.

Our scenario simulates the region of Daeyon-Busan in South Korea with 10 RSUs and 26 left-hand driving vehicles for 100 s with no pause time. Each vehicle transmits a 512 byte block header ten times/second to the RSUs at a rate of 6 Mbps. Here, RSUs acting as miners are placed 50 m apart to collects the data broadcasted and uploaded by vehicles. All numerical results were carried out using a computer desktop with Ubuntu OS version 16.04 that was installed on a virtual machine, Oracle VM VirtualBox. The computer has the following specifications: CPU Intel(R) Core(TM) i5-4690 CPU @ 3.50 GHz with 16.00 GB RAM. The simulation parameters are listed in Table 3, and the results are given in Figures 5–13.

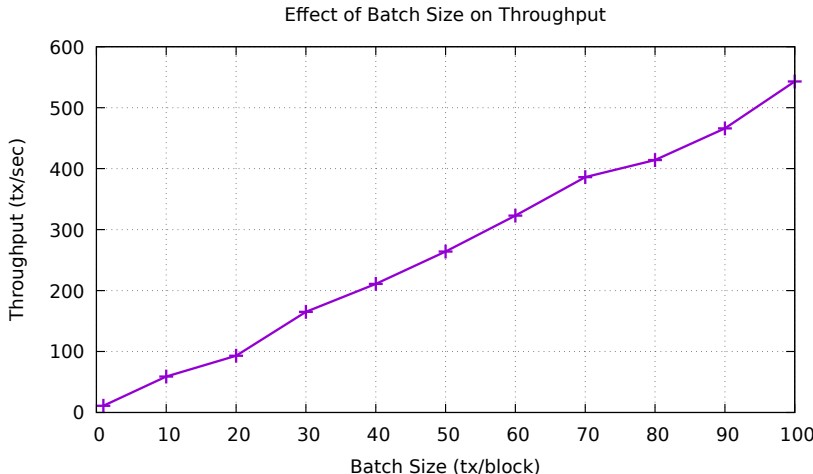

**Figure 5.** Effect of batch size on throughput.

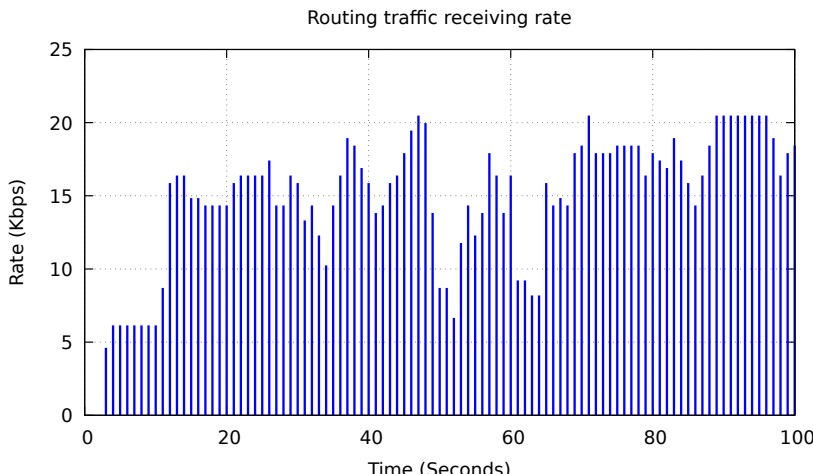

**Figure 6.** Routing traffic receiving rate.

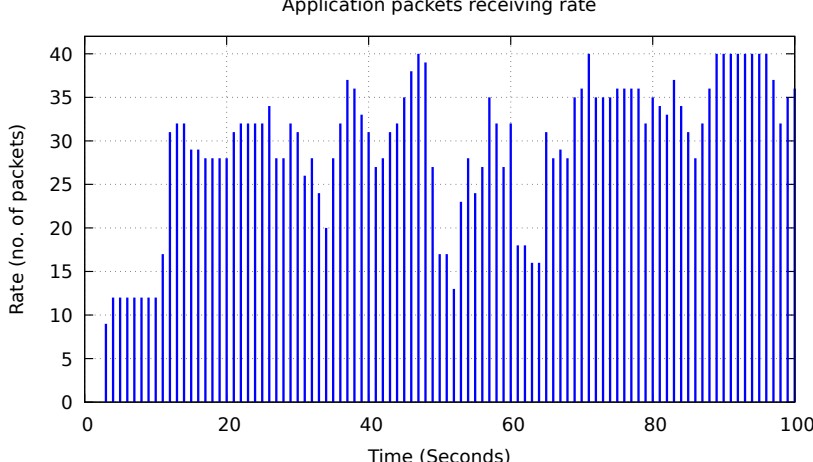

**Figure 7.** Application packets receiving rate.

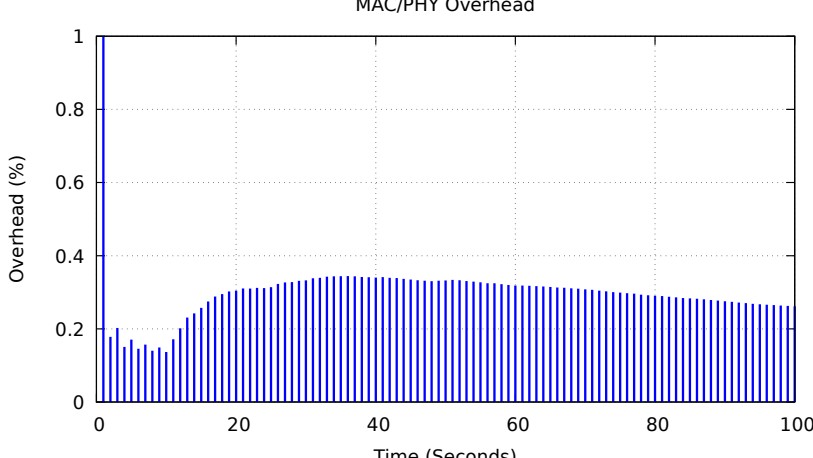

**Figure 8.** MAC/PHY overhead.

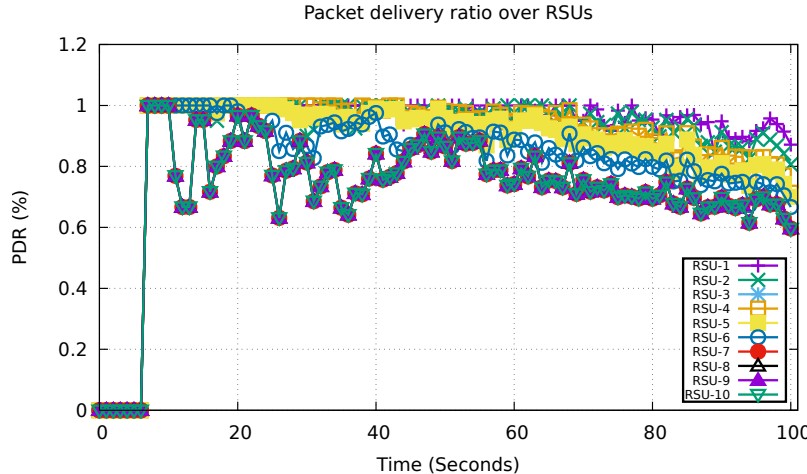

**Figure 9.** Packet delivery ratio over RSUs.

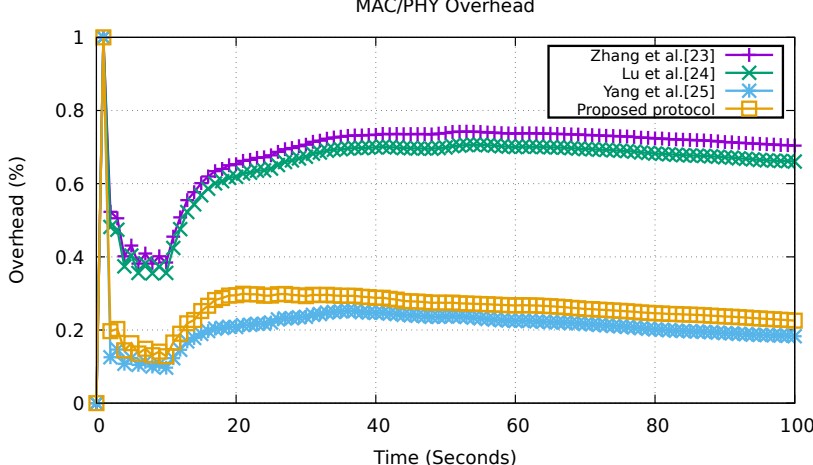

**Figure 10.** MAC/PHY overhead comparison.

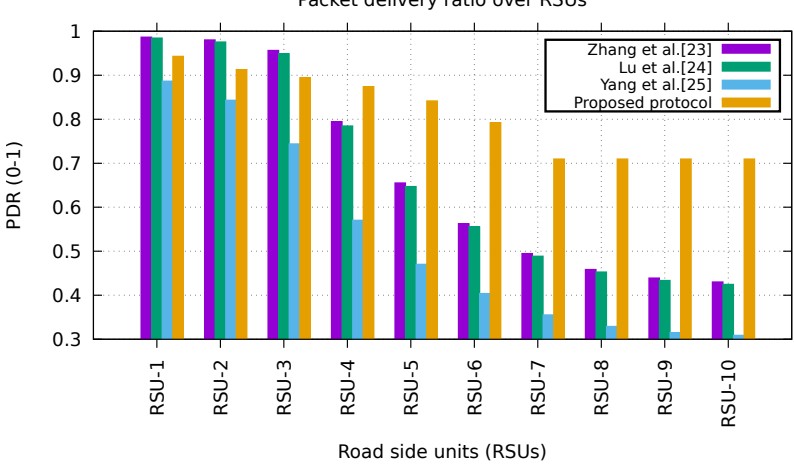

**Figure 11.** Packet delivery ratio over RSUs comparison.

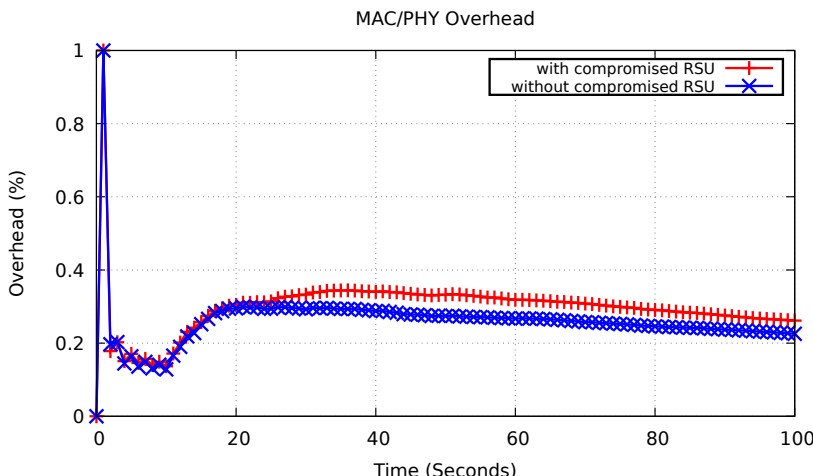

**Figure 12.** MAC/PHY overhead with compromised RSUs.

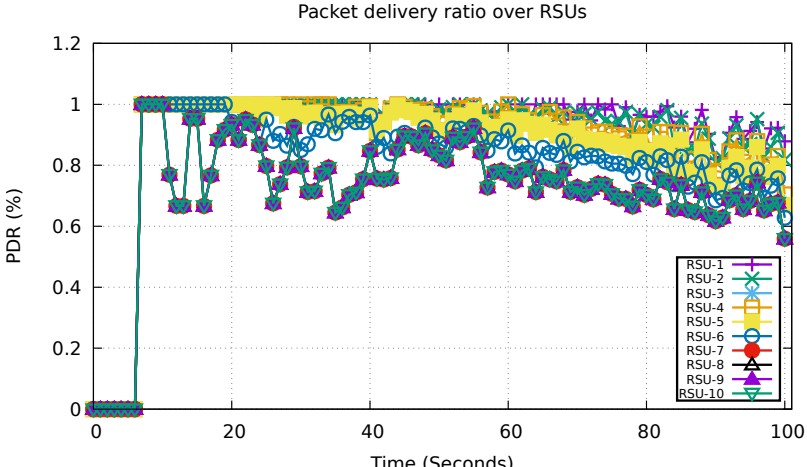

**Figure 13.** Packet delivery ratio with compromised RSUs.

**Table 3.** Simulation parameters.

| Parameter | Value |
|---|---|
| simulators | NS-3, SUMO, OSMWebWizard |
| simulation time | 100 s |
| MAC type | IEEE 802.11p |
| WAVE ITS band | 5.9 GHz |
| routing protocol | OLSR |
| physical mode | OFDM (6 Mbps rate) |
| fading model | Nakagami fading |
| propagation loss model | Two-ray ground |
| vehicle speed | 20 m/s |
| number of RSUs | 10 |
| distance between RSUs | 50:500 m |

**Table 3.** *Cont.*

| Parameter | Value |
|---|---|
| power transmission | 20 dBm |
| transmission rate | 2.048 Kbps |
| data rate | 6 Mbps |
| application packet size | 64-byte |
| blockchain packet size | 512-byte |
| packet interval | 100 ms |
| antenna height | 1.5 m |
| channel bandwidth | 10 MHz |

### 4.2.2. Block Generation

Data sharing flow begins with the vehicles sending or uploading all transactions to RSUs in the edge layer. RSUs use their transaction processors (i.e., LSSC and RPSC) to ensure the integrity of the transaction. Then, preselected RSUs conduct the consensus mechanism to generate the block (which consists of transactions) before storing it into the blockchain network. Here, the block size is equivalent to batch size in our simulation. The term batch size refers to the number of transactions that are in every block of the blockchain. We run multiple tests by changing batch size up to 100 tx/block to evaluate the effect of batch size on throughput. As can be inferred from Figure 5, at the total batch size of 100 tx/block, the throughput reaches 543 tx/s. The throughput is shown to increase linearly over different batch sizes.

### 4.2.3. Data and Application Rate

The data and application rate parameter refers to the total number of successfully sent vehicle data in a unit of time (in bit per second for data rate, and in the packet for application rate), which shows the transmission rate effectiveness in vehicular networks. It also represents an average traffic flow rate where the higher value indicates a better system performance [36]. All vehicles attempt to continuously route 64 byte application packets at a rate of 2.048 Kbps in our scenario. Figure 6 shows the application data receiving rate with the overall average throughput remaining at 140 Kbps, whereas Figure 7 shows the receiving rate at an average rate of 25 packets per second. We consider the application packet as well as the blockchain packet to evaluate the performance of application traffic in our system.

### 4.2.4. Mac/Phy Overhead

Medium access control and physical (MAC/PHY) layer overhead are essential factors that influence the vehicular network's performance. Generally, the broadcast channel is overwhelmed by the vehicle's authentication requests that may cause overload RSUs. Therefore, it affects the total average of throughput in the system and increases the vehicle's delay authentication. MAC/PHY overhead is calculated based on the below equation.

$$MAC/PHY_{overhead} = \frac{PHYBytes_{total} - AppBytes_{total}}{PHYBytes_{total}} \tag{3}$$

From the equation above, $PHYBytes_{total}$ and $AppBytes_{total}$ denote, respectively, the physical and data link layer traffic that represents the physical/user network layer in our proposed model architecture. On the other hand, the value of MAC/PHY overhead is in the range [0, 1] and is given by Equation (3). Here, the lower value indicates better performance of the system, and vice versa. Figure 8 shows the result of MAC/PHY overhead in 100 s simulation time. As shown in the figure, we can see that the overhead is relatively stable after 20 s of simulation time within the range of 0.2 to 0.3, and it even slightly decreases

over time. Based on these results, our proposed protocol is relatively efficient due to does not incur high overhead.

### 4.2.5. Packet Delivery Ratio (Pdr)

PDR is the ratio of the total number of packets successfully delivered in the destination nodes (RSUs) compared to the gross packet transmitted from the source node during the communication on the network. In this sense, the higher number of vehicle packet data reaching the RSU, the better performance of the system. In our scenario, PDR represents the blockchain packet broadcasted by vehicle to the RSUs. PDR was computed by calculating the ratio of the total number of received packets ($P_{receive}$) to the total number of sent packets ($P_{sent}$) as shown in the following equation.

$$PDR = \frac{\sum P_{receive}}{\sum P_{sent}} * 100 \tag{4}$$

Figure 9 shows the PDR ratio of our proposed protocol over 10 RSUs on different distances. In the beginning, the PDR value is low due to an increasing number of hops between vehicles and RSUs, while the protocol needs to complete a route-finding process before packets can be successfully transferred. The stable PDR results are obtained between RSU-1 and RSU-5, but a consecutive slump in the packet delivery between RSU-6 and RSU-10. As a result, we can see that RSU-1 has the highest PDR due to its nearest distance in this scenario. Besides, RSU-10 as the farthest distance has the lowest PDR caused by the fading channel's effect.

### 4.2.6. Comparative Analysis

We compare our proposed protocol with other protocols by Zhang et al. [10], Lu et al. [37], and Yang et al. [8]. We simulate all protocols using the parameter listed in Table 3 for the fairness comparison. In this scenario, MAC/PHY overhead and PDR are considered two parameters for analyzing each protocol's performance. Generally, the simulation results show that our proposed protocol performance outperforms all of the comparison protocols. Figure 10 shows that the overhead of our proposed is lower than the protocol in [10,37], while that in [8] is slightly better. Fortunately, we can see in Figure 11 that the proposed protocol PDR is more consistent along the time in varying transmission ranges rather than other protocols. Within the range where the delivery ratio was appreciable, there is adequate connectivity, assuring that every packet arrived in the RSUs. Even though the PDR of the protocols in [10,37] have appreciable values between RSU-1 to RSU-3, they then decrease linearly as distance increases on RSU-4 and so forth. Therefore, in the PDR context, our proposed protocol outperforms other protocols at different RSU locations.

Additionally, we also evaluate our proposed model in the case of attacks. In particular, we simulate the case of a number of malicious vehicles that compromise 33% total number of RSUs (to depict the maximum number of malicious nodes in PBFT). As illustrated in Figures 12 and 13, the result shows that there is a slight performance decrease for both the overhead and the PDR value. Nevertheless, in general, it can be inferred that there is no significant effect on system performance, even though there are 33% compromised RSUs in our proposed model. Thus, our proposed model can reach decentralized trust management in VECNets.

## 5. Discussion

Compared to the centralized approach of VECNets, our presented model using consortium blockchain performed adequately in security aspects, as shown in Section 4.1. Our system aims to avoid SPoF attacks as a bottleneck in a centralized framework that might expose users' private data due to compromised central data server. Contrary, the proposed system is controlled by multiple authorized RSUs, which will execute a consensus mechanism to reach an agreement before storing the data to the blockchain network. Thus,

the user's private information cannot be exposed throughout the data sharing process. Moreover, the cryptography algorithm guarantees the vehicle identity anonymity and communication security. In order to evade the malicious vehicle that may transmit false messages to the system (spoofing attacks), we offer a message credibility evaluation conducted by neighboring vehicles. The neighboring vehicle can comprehensively evaluate the transmitted message and generate ratings for the message's trustworthiness. Nevertheless, the malicious vehicle may indeed act as a neighboring vehicle and generate a false assessment by upload an unfair rating (false rating attacks) to RSU. Fortunately, due to the limited number of attackers and the guarantee of cryptography encryption on vehicle authentication, the unfair ratings might hardly alter the aggregated trust values in authorized RSU. Furthermore, a consortium blockchain with multiple preselected RSUs is utilized to prevent the compromised RSUs that might be creating and broadcasting modified blocks (data modification attacks) in the consensus process. Therefore, based on the discussion above, our proposed system can reach decentralized trust management in VECNets.

However, further discussion will be required in order to form a prevention approach rather than only a detection approach [38], such as to avoid attacker interference in transaction propagation time [39,40]. One of the solutions is by employing a robust message authentication mechanism to facilitate the vehicles data sharing process securely while preventing attackers from hijacking transactions, identifying the message's identity, and stopping the transaction's propagation of the system [41]. As pointed out in [14], there are several techniques can be leveraged to enhance the security and privacy of blockchain-based VECNets, such as group signature, homomorphic encryption, attribute-based encryption, and ring signature. In the context of the consortium blockchain framework, a group signature is more suitable since the signer's identity can be hidden among multiple users [14]. This algorithm allows all messages to be encrypted with the sending vehicle's private key before sending. The vehicle's anonymity is protected because no one knows to whom the message is sent. Thus, the attackers cannot expose transactions without cracking the sender's private key. Therefore, the more robust data authentication technique makes the system more secure from malicious vehicles.

## 6. Conclusions

We have presented a consortium blockchain and smart contracts to form distributed data storage and secure data sharing. A smart contract is empowered to enable decentralized trusted data sharing by leveraging LSSC and RPSC smart contracts of RSUs in the edge node layer. Therefore, RSUs act as data aggregators as well as miners in VECNets. We propose the vehicle's message reputation to tackle fake messages that may cause traffic congestion or even accidents. Consequently, all broadcasted messages obtain ratings from its neighboring vehicle to ensure the message's credibility. Here, PBFT is utilized to perform the consensus mechanism due to its merits, such as consistency, maturity, high efficiency, and small resource consumption, making it suitable for our proposed scheme. Moreover, PBFT allows the existence of strange nodes without affecting the consensus result. Our proposed scheme also enables vehicles with a good rating that generates the correct messages and contributes to the trust value assessment to obtain certain rewards. Thus, they will be motivated to be honest to share their data in the vehicular network. Numerical results show that our proposed protocol outperforms in increasing the general performance parameter on vehicular networks. In our future work, besides improving the performance of other system metrics and implementing a vehicle authentication mechanism, we also consider integrating artificial intelligence (e.g., deep learning) into the blockchain platform in the VECNets framework.

**Author Contributions:** Conceptualization, M.F.; methodology, M.F.; software, M.F.; validation, M.F.; formal analysis, M.F.; investigation, M.F., K.-H.R.; resources, K.-H.R.; data curation, M.F.; writing—original draft preparation, M.F.; writing—review and editing, K.-H.R.; visualization, M.F.; supervision, K.-H.R. Both authors have read and agreed to the published version of the manuscript.

**Funding:** This research was supported by the MSIT (Ministry of Science and ICT), Korea, under the ITRC (Information Technology Research Center) support program (IITP-2020-0-01797) supervised by the IITP (Institute of Information and Communications Technology Planning and Evaluation). This research was supported by the Republic of Korea's MSIT (Ministry of Science and ICT), under the High-Potential Individuals Global Training Program) (2020-0-01596) supervised by the IITP (Institute of Information and Communications Technology Planning and Evaluation).

**Institutional Review Board Statement:** Not applicable.

**Informed Consent Statement:** Not applicable.

**Conflicts of Interest:** The authors declare no conflict of interest.

## Abbreviations

The following abbreviations are used in this manuscript:

| | |
|---|---|
| VANET | Vehicular Ad hoc Network |
| ITS | Intelligent Transportation System |
| VECNet | Vehicular Edge Computing Network |
| PBFT | Practical Byzantine Fault Tolerance |
| MEC | Mobile Edge Computing |
| OBU | Onboard Unit |
| RSU | Roadside Unit |
| TA | Trusted Authority |
| V2V | Vehicle to Vehicle Communication |
| V2I | Vehicle to Infrastructure Communication |
| DSRC | Dedicated Short-Range Communication |
| LSSC | Local Storage Smart Contract |
| RPSC | Record Pool Smart Contract |
| SPoF | Single Point of Failure |
| SUMO | Simulation of Urban Mobility |
| WAVE | Wireless Access for Vehicular Network |
| OLSR | Optimized Link-State Routing |
| MAC/PHY | Medium Access Control/Physical Layer |
| PDR | Packet Delivery Ratio |

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
