# Peer review of "On Blockchain-Enhanced Secure Data Storage and Sharing in Vehicular Edge Computing Networks"

_applsci, doi:10.3390/app11010414_

Round 1
Reviewer 1 Report
The paper seems to propose using blockchain technology along with smart contracts to secure communications in VECNets (VANETs at the Edge).
I intentionally use word "seems" for two reasons: first, written english is so poor that it is extremely difficult to follow the narrative and fully understand the content, second, the evaluation section only cares about network performance, not security. Most of security aspects are simply and briefly discussed in text, and seems to exclusively stem from the blockchain nature, not from the authors' contribution. Also, crucial information is missing about the authors' implementation: is it available for testing? is it entirely from scratch or does it rely on existing infrastructure?
Besides these fundamental flaws, there are several sentences in the paper that either require further discussion, or I cannot simply agree on. For instance (authors' text in italics, reviewer's emphasis in bold):
- "where the edge node is preselected [5] that has the responsibility to execute the consensus process and determine the generation of each block" how is it preselected?
- "data in the blockchain are consistent, reliable, timely, and widely accessible." data is not at all timely on most blockchains, as the consensus protocol introduces notable overheads.
- "we use the PBFT algorithm to implement in consortium blockchain due to the requirement of the real-time transaction of VECNets" I think most researchers and practitioneers in either blockchain or real-time systems would argue that PBFT and real-time cannot coexist in the same sentence...
- "the receiving vehicle on a certain distance and time of the occurring events is designed to aggregates the information messages. Then, they generate the rating based on the message’s credibility" how much distance? how aggregation is performed? how is the message credibility computed? is it a simple majority vote on received messages as described later on the manuscript?
- "α and β are two predefined parameters which adjust the rate of the message credibility" how are these set?
- "PBFT is optimized because it is suitable to cope with the requirement of the system" this sentence needs an in-depth discussion, as it is not clear at all what "optimized" means in this context...
- "we assume that Vs provides a valid message about a particular event in the regional A with a specific time" what if Vs is malicious?
As it stands, the manuscript has too many flaws among which the serious evaluation problem mentioned at the beginning to be considered for publication.
Reviewer 2 Report
The article has a good level and structure, brings a clearly presented methodology and its own solution. The results are presented in a form that requires some knowledge in the issue, for the less knowledgeable, this form of presentation is more difficult to understand.
I recommend defining the abbreviations used when they first appear in the text
Figure 4 - the texts in the image are written in too small a font - they are unreadable at normal display size
At the end, it would be good to add information on whether the research will be continued, if so in what direction? Are the authors considering using the results in practice?
Reviewer 3 Report
In this paper, the authors propose7consortium blockchain and smart contracts to ensure a trustworthy environment for secure data8storage and sharing in the system.
Paper overall is well organized, and covers an interesting topic
I suggest the following revisions
- English spelling and grammar should be checked thoroughly.
- The Figures in the experiments should be redone, they seem low quality and blurry and also the manner in which the Legend is placed make the figures seem fabricated. Please fix
- I think in general the results and comparative analysis should be presented in a better manner, better description and connections between text and figures is needed.
- Please find 5-10 recent papers from MDPI papers in the scope of your work and and add to the related work and references. Sensors MDPI would be a good place to start looking.
- Notation table can appear earlier in the text
- More of an in depth security analysis and attack analysis is needed, see
Reviewer 4 Report
This paper proposes a new architecture design based on the use of a blockchain in vehicular edge computing networks which includes incentives for vehicles to contribute and share their data.
The authors are encourage to review the following items:
In abstract:
- PBFT is not explained nor justified
In introduction section:
- The table 1 it is placed in the wrong page since it is only referenced in page 4.
- The figure 1 it is placed in the wrong page since it is only referenced in page 5
In Proposed Architecture section:
- This section is highly segmented in subsections. An example is the section 3.2.2 and that has only a paragraph and 3.2.3 includes only a sentence.
- Figure 3 and 4 could have better detail (use vector graphics?)
In Security Analysis and Performance Evaluation section:
- all figures could be improved in terms of resolution (use vector graphics?)
Round 2
Reviewer 1 Report
The authors addressed many of my initial concerns and most of all sensibly improved English written form.
I still think authors need to rework the evaluation section as per my previous report, as again it is much more focussed on evaluation of the system performance rather than of security aspects. For instance, they state "Nevertheless, the malicious vehicle may indeed act as a neighboring vehicle and generate a false assessment by upload an unfair rating (false rating attacks) to RSU. Fortunately, due to the limited number of attackers and the guarantee of cryptography encryption on vehicle authentication, the unfair ratings might hardly alter the aggregated trust values in authorized RSU". Assessing through attackers simulations what would happen in similar cases would add value to the manuscript.
Another aspect that I think is still not clear is which kind of blockchain technology has been used "off the shelf" or implemented as part of the simulations: NS3 is a discrete event network simulator, not a blockchain, and SUMO is a traffic simulator, so again not a blockchain. Where is the blockchain in your simulations?
The latter issue is particularly urgent to address in my opinion, as it undermines the whole evaluation section.
Reviewer 3 Report
Authors have handled my suggested revisions properly.
I think the Related Works can be better presented. Perhaps pick 5 closest related papers and discuss in depth??
Author Response
The authors thank the reviewer for useful feedback.
We have added Subsection 2.4 Related Work and discuss several closest related papers. The revision can be checked in Line 183-213.